# Hydrolytic Degradation of Polylactic Acid Fibers as a Function of pH and Exposure Time

**DOI:** 10.3390/molecules26247554

**Published:** 2021-12-13

**Authors:** Radhika Vaid, Erol Yildirim, Melissa A. Pasquinelli, Martin W. King

**Affiliations:** 1Fiber and Polymer Science Program, Wilson College of Textiles, NC State University, Raleigh, NC 27606, USA; rvaid@ncsu.edu (R.V.); melissa_pasquinelli@ncsu.edu (M.A.P.); 2Department of Chemistry, Middle East Technical University, 06800 Ankara, Turkey; erolyil@metu.edu.tr; 3Department of Forest Biomaterials, College of Natural Resources, NC State University, Raleigh, NC 27606, USA; 4College of Textiles, Donghua University, Shanghai 201620, China

**Keywords:** polylactic acid, hydrolytic degradation, bioresorbable polymers, ReaxFF, pH, sustainable materials, fibers

## Abstract

Polylactic acid (PLA) is a widely used bioresorbable polymer in medical devices owing to its biocompatibility, bioresorbability, and biodegradability. It is also considered a sustainable solution for a wide variety of other applications, including packaging. Because of its widespread use, there have been many studies evaluating this polymer. However, gaps still exist in our understanding of the hydrolytic degradation in extreme pH environments and its impact on physical and mechanical properties, especially in fibrous materials. The goal of this work is to explore the hydrolytic degradation of PLA fibers as a function of a wide range of pH values and exposure times. To complement the experimental measurements, molecular-level details were obtained using both molecular dynamics (MD) simulations with ReaxFF and density functional theory (DFT) calculations. The hydrolytic degradation of PLA fibers from both experiments and simulations was observed to have a faster rate of degradation in alkaline conditions, with 40% of strength loss of the fibers in just 25 days together with an increase in the percent crystallinity of the degraded samples. Additionally, surface erosion was observed in these PLA fibers, especially in extreme alkaline environments, in contrast to bulk erosion observed in molded PLA grafts and other materials, which is attributed to the increased crystallinity induced during the fiber spinning process. These results indicate that spun PLA fibers function in a predictable manner as a bioresorbable medical device when totally degraded at end-of-life in more alkaline conditions.

## 1. Introduction

Polymers from bio-based resources have become important materials for many applications in the biotechnology, medical device, and pharmaceutical industries, as well as for applications in green chemistry and engineering, such as packaging. Their unique properties, such as biocompatibility, biodegradability, and bioresorption, have led to their extensive usage, for example, in treating injuries and diseases through tissue engineering, wound dressings, and drug delivery [1,2,3,4,5,6,7,8,9,10,11,12,13]. With new discoveries in the field of polymers and biomedical applications, it has become critical to have a rapid and reliable methodology to evaluate and shortlist the polymer candidates based on their degradation performance, particularly for implantable devices, as it affects in vivo behavior of polymers once placed inside the body.

Poly-l-lactic acid (PLLA) and poly-d-lactic acid (PDLA) are formed after polymerization of optically active l-lactic acid and d-lactic acid monomers respectively and the polymers are semi-crystalline in nature. l-lactide is a naturally occurring isomer and is more readily available compared to its counterpart D-lactide. However, commercial PLA polymers obtained from a natural source, such as Natureworks’ Ingeo polymer, which is derived from corn oil, contain both isomers with about 98% PLLA and about 2% PDLA. PLLA generally has around 37% crystallinity, which is dependent on the molecular weight and processing parameters. It has a higher Tg of about 60–65 °C, and its Tm is approximately 175 °C. In addition, PLLA exhibits high tensile strength (~59 MPa) and high modulus (~3.8 GPa) [9,10,11]. It has good tensile strength, low extension, and a high modulus, which makes it a suitable candidate for many applications, especially in load bearing ones such as surgical sutures and orthopaedic fixtures. Some of the commercially available PLLA based orthopaedic products are Phantom Suture Anchors (DePuy, Raynham, MA, USA), Full Thread Bio Interference Screws (Arthrex, Naples, FL, USA) and Meniscal Stingers (Linvatec, Lago, FL, USA). Superior mechanical properties have resulted in the use of PLLA as textile scaffolds for ligament replacement and augmentation devices to replace nondegradable fibers, such as Dacron polyester. Biomedical research continues on PLA as the base material for blood vessel conduits and treatment for lipoatrophy [12,13].

Furthermore, PLLA is a slower degrading polymer compared to other aliphatic polyesters like polyglycolic acid (PGA). This is attributed to the voluminous methyl groups present in the backbone chain that are responsible for its initial hydrophobic surface, which delays initial hydrolytic degradation. Higher molecular weight PLA implants have been observed to remain from two to over five years before they are totally resorbed in vivo [9,10]. In addition, PLA bioresorption tends to take place through bulk erosion [10]. Nevertheless, the rate of degradation depends to a large extent on the degree of crystallinity of the material as well as the porosity and thickness of the implant. In aqueous environments, bio-based polymers tend to lose some of their strength and elastic modulus in the first six months due to the plasticizing action of water, and thus no significant changes will occur in the mass until much later. Therefore, currently copolymers of L-lactides or D-lactides with glycolides are being studied with the objective of developing polymers with a wider range of properties and thereby increasing their potential range of application [9,10]. In addition, since hydrolytic degradation is dependent on the surrounding environment, pH plays a pivotal role in determining the rate of PLA degradation. 

The goal of this work was to investigate the degradation characteristics of PLA fibers as a function of a wide range of pH values and times of exposure. In tandem with experimental characterization, molecular dynamics (MD) simulations using a reactive force field and density functional theory (DFT) calculations provided molecular-level characteristics of PLA degradation that may have impacted its properties and performance when used over time as a medical device in the body or in green chemistry and engineering applications, including end-of-life total resorption. 

MD simulations numerically solve Newton’s equations of motion (F = ma; where m is the mass of particle and a is the acceleration attributed to the displacement) for all (atomic) particles in the system to predict equilibrium and dynamic properties of complex molecular-level systems that cannot be calculated or observed analytically [14,15,16] Classical MD simulations typically use force fields to represent the intermolecular and intramolecular interactions among the atoms within the system, and use thermodynamic ensembles, such as the grand canonical, for statistical control [17,18]. DFT calculations quantify quantum level interactions and reactivity, and are based on following the Hohenberg-Kohn theorem, where the total energy of the system is given as a function of the electron density [19,20]. These molecular modeling tools are excellent for green chemistry applications when used in tandem with experimental investigations, and have been used in polymer science to investigate a variety of molecular features, such as the effect of temperature, additives, fillers, and solvents on polymer structures and their dynamics, as well as probe the effect of changes in conditions and environment on the molecular system and material properties [21,22,23,24,25,26,27,28,29,30,31]. For example, a change in bond breakage/degradation in response to increased temperature in MD simulations results in the deterioration of mechanical properties of polymers that can be predicted over space and time. The framework from this study can also be used to advance the development of structure-process-property relationships for bio-based fibers [32].

## 2. Materials and Methods

### 2.1. PLA Fibers and Buffer Solutions

The PLA resin (6100 HP) supplied by NatureWorks™ (Minnetonka, MN, USA) was used as the starting material. This particular grade of PLA contains greater than 98% L-isomer and a small amount of D-isomer. PLA melt spinning, and drawing was conducted using the Hills multifilament research spinning line at The Nonwovens Institute, North Carolina State University. A spin-pack with a 69-hole spinneret was used, resulting in a multifilament yarn with 69 filaments. The spinning was carried out at 220 °C at a winding speed of 1500 m/min. Following this, drawing was carried out at 75 °C with a draw ratio of 2.8. Buffer solutions at pH 2, 3, 7.4 and 10 were purchased from VWR.

### 2.2. Experimental Degradation Studies

Degradation was carried out at pH 2, 3, 7.4 and 10 for 5, 15 and 25 days at 37 °C. These pH conditions were selected to mimic the pH conditions prevailing at various locations in the human body, although such a wide pH range also exists in different soils and environmental conditions. The temperature of the study, 37 °C, is normal body temperature. However, PLA tends to have slow degradation rates when implanted in vivo, at a temperature of 37 °C and a pH of 7.4, demonstrating complete degradation and total loss of mass over 24–30 months. For this study 25 days was chosen as the exposure time, as the focus of this study was to understand the molecular-level details of strength loss at various pH conditions, not to monitor the complete degradation or resorption process. 

### 2.3. Measurement of Mass Loss

The degraded samples were measured for changes in mass before and after degradation on an Ohaus Adventurer Pro Precision balance. This was done to register degradation with respect of loss of mass over time in different pH conditions, which also play a critical role in the change in tensile properties and hence will affect the properties of polymeric and fiber implants placed in-vivo. The percentage mass loss (*WL%*) is calculated by Equation (1),
(1)$$WL\%=(\frac{w_{o}-w_{f}}{w_{o}})\;\times \;100$$
where *W_o_* and *W_f_* are the respective initial and the final mass of the PLA fiber specimen.

### 2.4. Tensile Testing

Tensile testing was performed on an XQ-1C Fiber Tensile Tester using ASTM standard D3822 with a gauge length of 1 mm using a 2 N load cell. The speed of testing was kept at 15 mm/min for all the specimens of the different samples analyzed over different times of degradation and pH conditions. For each sample, triplicate measurements were taken, and 15 specimens were analyzed for each case.

### 2.5. Scanning Electron Microscopy (SEM)

An SU8010 Hitachi scanning electron microscope was used for analyzing the degradation of the PLA fibers at different pH conditions after 25 days. This was undertaken at Donghua University, Songjiang campus, Shanghai. The fiber specimens were attached to SEM stubs with carbon tape and coated with gold in order to make them conductive and reduce the chance of charging under the electron beam. SEM was performed to observe any changes in the topographical features of the fibers after exposure to different pH conditions. It was hoped that the observations would point to the degradation mechanism that is relevant for in vivo polymer applications. 

### 2.6. Crystallinity Measurement

An Rigaku SmartLab X-ray diffractometer (XRD) was used to measure the change in % crystallinity and crystal size of PLA fibers on degradation in different pH conditions after 25 days. The XRD scan was executed from 5–45°, with a step of 0.05°. PDXL software was used to undertake baseline correction and peak fitting calculations so as to identify the amorphous regions. Peak heights and area ratios were calculated from peak deconvolutions using OriginPro software (OriginLab Corporation, Northampton, MA, USA) [33].

### 2.7. MD Simulations

Molecular models were built, and MD simulation trajectories were analyzed using MAPS 4.2 by Scienomics [34], and the MD simulations were performed using LAMMPS [35]. The MD simulations were designed to investigate polymer/water systems at different concentrations of hydrogen (H^+^) and hydroxyl (OH^−^) ions to simulate different pH conditions and to predict the changes in molecular weight [23,24,25,26,27,28,29,30,31,32,33]. The number of H^+^ and OH^−^ ions was fixed at 10, 30, 50 and 100 in the individual systems, and the neutrality of the system was established by adding sodium (Na^+^) and chloride (Cl^−^) ions. The systems were built using an amorphous cell module with 20 polymer chains, each with 20 repeat units, and 250 water molecules. In addition to the different ion concentrations, the control system did not include any ions. Additionally, systems of just one polymer chain with 20 repeat units were also created to facilitate the analysis of the reaction products at the molecular level. Three systems were built for each composition for statistical analysis. Each molecular model system was equilibrated to minimize energy and optimize density, and the MD simulations were performed with the canonical ensemble which maintains a constant number of particles, volume, and temperature (NVT), using the Nose-Hoover thermostat at 37 °C using ReaxFF for aqueous systems [36]. (A.C.T. Van Duin provided the ReaxFF reactive force field software). Changes in molecular weight, reaction products, and rate of degradation were obtained through analysis of the simulation output. 

To extract the rate constant of degradation of PLA from the MD simulation trajectory, the rate equation reported for autocatalytic degradation was used (Equation (2)), where the change in concentration of acid end groups [COOH] is determined by the change in concentration of water [H_2_O], ester [E] and the acid generated.
(2)$$\frac{d\left[\mathrm{COOH}\right]}{\mathrm{dt}}=k\left[\mathrm{COOH}\right]\left[H_{2}O\right]\left[E\right]$$

[COOH] and [E] can be related to the number average molecular weight (Mn) using Equations (3) and (4) [37].
(3)$$\left[\mathrm{COOH}\right]=\frac{\rho }{\mathrm{Mn}}$$
(4)$$\left[E\right]=\frac{\rho }{\mathrm{Mn}}\left(2\mathrm{DP}-1\right)$$

In these equations, ρ is the density of the polymer sample (about 1.2 g/cm^3^) and DP is the average degree of polymerization, defined as the ratio Mn/M, where M is the molecular weight of the repeating unit, equal to 72 g/mol for PLA. By substituting the values for [COOH] and [E] into Equation (2) and assuming the value of DP as mentioned, the rate constant was calculated using Equation (5).
(5)$$\;\frac{\mathrm{dLN}\left(M_{n}-M\right)}{\mathrm{dt}}=-k$$

The number average molecular weight with respect to carbon (C_n_) was considered in place of M_n_ to negate any effects of molecular weight due to attachment of Na^+^ and Cl^−^ to the polymeric systems in the MD simulations. The weight of the repeat unit was replaced by the weight of carbon in the repeat unit (C_nr_) (36 g/mol for PLA). In addition, as the degradation was being evaluated any higher molecular weight intermediates formed in MD simulations were neglected. The number average molecular weight with respect to the number of carbon atoms (C_n_) in the system was calculated over the simulation trajectory. Thus, using Equation (6), the rate constant (k) was calculated for the degradation in the system.
(6)$$k=-\frac{\mathrm{dLN}\left(C_{n}-C_{\mathrm{nr}}\right)}{\mathrm{dt}}$$

### 2.8. Density Functional Theory (DFT) Calculations

DFT calculations were performed with Gaussian 16.A03 [38] to determine the reactivity of PLA with five repeat units and water as an implicit solvent. A B3LYP/6-31+g(d) level of calculation [39] was used to calculate geometry optimizations, molecular electrostatic potentials (ESP), as well as surface and atomic charges. Additionally, a point in the space around a molecule gives an indication of the net electrostatic effect, which contributes to the total charge distribution (electron + nuclei) of the molecule and correlates with dipole moments, electronegativity, partial charges, and the chemical reactivity of the molecule. This also provides a visual depiction of the relative polarity of the molecule with regions for susceptible electrophilic and nucleophilic attack (NPA) at various reactive sites. The Fukui function was thus implemented by adopting the finite-difference approach of using atomic charges based on NPA calculations, since, unlike Mulliken population analysis, electron densities depicted from NPA analysis are reported to be robust and reliable. By default, a whole electron was either added or removed. However, the calculation was not restricted to this amount. The Fukui function for electrophilic attack ($f^{-}$) is given when 1 electron is removed [39,40,41] as,
(7)$$f^{-}=\rho \left(N\right)-\rho {\left(N-1\right)}^{\;}$$
where N and ρ represent the number of electrons and density of electrons. Equally, the Fukui function for the nucleophilic attack ($f^{+}$) is given when 1 electron is added, as given by,
(8)$$f^{+}=\rho \left(N+1\right)-\rho \left(N\right)\;$$

Further, the dual descriptor (f(r)) is a combination of the two Fukui functions for electrophilic and nucleophilic attack having a positive value where it is electrophilic (attracting electron rich or donating species or nucleophiles) and negative where it is nucleophilic (attracting electron deficient or withdrawing species or electrophiles). This dual descriptor is implemented as the difference between the Fukui plus and Fukui minus functions, as given in Equation (9).
(9)$$\;f\left(r\right)=\frac{1}{2}\;\left(f^{+}+f^{-}\right)$$

## 3. Results

### 3.1. Effect on Mass

Figure 1 illustrates the effect of pH exposure over 25 days on the mass of the fiber samples and indicates that the mass loss increased when the PLA fiber was subjected to alkaline conditions of pH 10, in contrast to acidic and neutral pH conditions. In addition, the mass loss was observed to gradually increase from 5, to 15 and 25 days when at pH 10, unlike at lower and neutral pH conditions where no significant change was observed over the 25 days. Figure 1a indicates that at pH 10, 0.2% mass loss occurred for the PLA fibers after five days, which increased 2.9 times to 0.6% after 25 days. Furthermore, Figure 1b provides the percentage of mass retention and indicates that there was no significant change in mass at the lower pH values, whereas at pH 10 it increased continuously over time. Similar results were obtained by Xu et al. [42] who observed a faster decrease in thickness of PLA fibers when exposed to alkaline conditions for extended times.

### 3.2. Effect on Mechanical Properties

Figure 2a contains the load-elongation curves for the PLA fibers, and demonstrates that the loss of peak load occurs after degradation at a range of different pH conditions. There is an initial increase in percentage elongation between five to 15 days. However, after 25 days the decrease is highest in alkaline conditions. As measured from triplicate samples with 15 specimens in each, the tensile strength in Figure 2b,c is observed to be reduced by 25% after five days and 40% after 25 days when exposed to pH 10. A similar decrease was observed at pH 3 and pH 7.4 over the first five days. Thereafter, the results levelled off with no significant change over the remaining time. However, at pH 2, a 10% loss in strength was detected after five days, and this loss increased to 25% after 15 days where it remained constant until Day 25. In Figure 2d the Youngs modulus of each PLA fiber did not follow an obvious trend. During the initial 15 days, there was a decrease noticed at pH 2, 3 and 7.4 followed by a minor increase after 25 days, which was not significantly different from the starting modulus. However, at pH 10, a significant increase in the Youngs modulus was detected after five days. Thereafter, unlike that of other pH’s, the tensile strength of the PLA fibers gradually decreased from five to 25 days. 

### 3.3. Effect on Surface Properties

Figure 3 provides SEM images of the surface and cross-sections of the PLA fibers after 25 days, which illustrates the topographical changes as a function of pH. The surface and cross-sectional views of the samples reveal that PLA degradation after 25 days at 37 °C was more rapid at pH 10 compared to the lower pH values of 2, 3, and 7.4. Striations and pores were observed on the surface of the PLA fibers after 25 days of degradation at pH 10. At lower pH values only striations were observed on the PLA fiber surface. Furthermore, from the cross-sectional views after 25 days at different pH conditions, no discernable pores were observed, unlike the surface of the PLA fibers exposed to pH 10.

### 3.4. Effect on Structural Properties

Figure 4 contains the XRD plots for the PLA fibers after 25 days of degradation. In Figure 4a, the prominent crystalline peak at ~16.4° was observed for all PLA fibers. However, it became narrower and sharper compared to the control after degradation. In Figure 4b, the crystalline peak at ~28.8° became narrower and sharper after 25 days of degradation at all pH values except for the control sample, where it can only be detected as noise. The peak height ratio for these two peaks ranged from 65–70 (Appendix A), with the peak area ratio being lowest for the sample degraded for 25 days at pH 10. Further analysis of Figure 4c revealed the presence of a low intensity peak at 33.3°, and due to its low intensity, the peak intensity could not be measured even after deconvolution. As a result the ratio for this peak in comparison to 16.4° could not be determined. These observations indicate that the level of crystallinity increased significantly on degradation at different pH conditions as compared to the control sample.

Finally, the % crystallinity and crystal size were calculated from the XRD plots and the highest intensity peak, respectively, and are given in Figure 4d. The % crystallinity increased by 35–37% after 25 days of degradation with a 90% maximum crystallinity value observed for the sample degraded at pH 10. Similarly, the highest crystal size was also observed for the sample degraded at pH 10. The crystallinity and crystal size for samples degraded at pH 2, 3 and 7.4, though significantly different from the control sample, were not observed to have much variance. 

### 3.5. Details of Degradation at the Molecular Level

Figure 5 contains a schematic of the various steps that are involved in the alkali assisted hydrolytic degradation of PLA, as observed from the MD simulations on both a single PLA chain and 20 chains in an aqueous environment. In the reaction mechanism, electrons being withdrawn are illustrated by red arrows, whereas blue arrows illustrate the donation of electrons, and green arrows signify the alkali attack. As expected based on what is reported in the literature [37], the hydrolytic degradation occurs at the ester linkages, and one hydroxyl and one carboxyl end group is generated. These observations validate using ReaxFF for aqueous environments during the MD simulations. In an aqueous environment, the methyl group at the α carbon acts as an electron donating group, making it more electronegative. Simultaneously, the oxygen at the carbonyl carbon is an electron withdrawing group, thus resulting in the higher negative charge concentration on it. This makes carbonyl carbon at the ester linkage electropositive with two electron withdrawing groups attached to it, resulting in a nucleophilic center susceptible to having an attack by the hydroxyl group in an alkaline system. This results in the formation of a tetrahedral intermediate, as observed from the MD simulations. 

In the presence of alkali, this process was observed in the MD simulations to be enhanced since the hydroxyl attack was initiated easily as compared to the hydronium ion attack. In the case of acid assisted hydrolysis, the hydronium ion is formed when the water and hydrogen ion react. Furthermore, assisted hydrolytic degradation by the direct attack on the ester carbon was observed in excess of OH^−^, which is sterically more hindered for water due to the presence of the methyl group. As this is a reversible process, some chains during the simulation trajectory return to the original polymer structure. However, in the presence of higher amounts of OH^−^, the ether connected to the tetrahedral intermediate (CH_3_CO^−^) causes hydrolysis of the PLA chain through the generation of oligomers with alcohol and carboxylic acid end groups, resulting in chain scission and hence polymer degradation. 

Analysis of the MD simulation for a single aqueous chain at 37 °C indicates that the onset of hydrolytic degradation is observed sooner in the presence of hydroxyl ions, unlike the neutral and acidic environments, where no or very little degradation was observed respectively, even at the end of the simulation. Even when intermediates were formed during the simulation trajectory (including ones with strong associations with the counterions, Na^+^ and Cl^−^), they were observed to not persist and tended to recombine into polymer chains. Furthermore, it was observed from these equations that there are only a few degraded species present in neutral and acidic environments in contrast to a larger number present in increasing alkaline conditions. These findings were also observed in the systems with 20 chains, where about 25% faster degradation was observed compared to single chain systems, particularly for alkaline conditions. The plot of the change in carbon molecular weight versus degradation time for the 20 chain system is given in Figure 6a. The slope was then calculated from the line of best fit through the average values given in Figure 6b to obtain the rate constants (k), which are given in the inset. Negative values signify degradation in the system as the longer chains break into smaller fragments. The highest extent of degradation takes place in the extreme alkaline conditions in the system with 100 OH^−^ ions with a k value of 2.43 ns^−1^. Almost horizontal lines were obtained for the other systems, especially for water and acidic systems, due to the lack of degradation during the simulation time period. These results corroborated the findings from the experiments where the highest degradation was observed under alkaline pH conditions.

DFT calculations were performed to validate the findings from MD simulations and those reported in the literature in terms of reaction pathways and degradation products for the hydrolysis of PLA. Figure 7a contains the optimized structure as well as the Fukui reactivity indices for the potential nucleophilic (f^+^), electrophilic (f^−^), and radical addition (f(r)) sites obtained based on the NPA atomic charges, where higher values of reactivity are shown in red and lower reactivities in green. These results indicate that the most susceptible atom for nucleophilic attack (f^+^) by the hydroxide anion is C31, the carbonyl carbon, whereas the α carbon to the ester linkage has a negative potential (green). These observations support the proposed degradation mechanism at higher pH values as discussed above. In terms of electrophilic attack (f^−^), the end groups of the PLA chain exhibit higher reactivity and thus have a chance of being attacked by H^+^ or other cations, which corroborates the observation of Na+ terminated intermediates in the MD simulations. Moreover, the 3D electrostatic potential surface diagram in Figure 7b, illustrates that a lower electron density (blue hue) is observed for the ester carbons (Appendix A). 

## 4. Discussion

PLA degradation has been widely investigated for diverse applications. The focus for biomedical applications has mainly been on the evaluation of hydrolytic degradation triggered by an autocatalytic mechanism that generates carboxylic acid and leads to bulk erosion of the PLA material. The range of pH considered in this study as a function of exposure time has, to our knowledge, not been investigated previously for PLA in fibrous form. Furthermore, this experimental work has been complemented with molecular modeling to provide molecular-level details of the effects of the extreme pH conditions. Specifically, the current study focuses on PLA fiber degradation at pH 2, 3, 7.4 and 10 to determine the mechanical performance, resorption profile, and surface properties after five, 15 and 25 days of exposure. The samples retrieved after in vitro degradation were collected and compared with the Day 0 control. The samples were characterized and analyzed by weight loss measurements, mechanical testing, SEM and XRD. Only a 25-day study was undertaken since the focus was on understanding the molecular-level mechanism that leads to strength loss in PLA fibers rather than on the complete degradation process. 

The analysis of the mass loss for PLA fibers given in Figure 8 reveals a gradual loss over the period of 25 days in alkaline conditions, particularly at pH 10. A gradual loss in the mass indicates surface erosion, which is highest under alkaline conditions. Furthermore, the strength loss was observed to be greater under alkaline conditions and reached 40% during the 25 day study. The strength loss, however, was limited to only 25% under acidic and neutral conditions. In surface erosion, strength loss is reported to occur after mass loss, which is contradictory to the observation made in this study. However, this can be due to the fact that the true area of the sample decreased with degradation, whereas the stress measurements are based on the original area calculated for the control sample. Thus, the ultimate load is calculated over the larger area, resulting in lower peak stress and strength values which is opposite to the anticipated trend. Also, due to the heterogenous loss in mass, the pores and striations are created on the fiber surface as observed by SEM (Figure 3), resulting in weak points in the fiber that contribute to the loss in strength during degradation under alkaline conditions.

Further confirmation that these PLA fibers experienced surface erosion is given in Figure 9 which contains magnified SEM images that illustrate the presence of pores on the surface but not in the core. This is in contrast to what has been observed previously in other PLA systems. The occurrence of elliptical pores (average values of a = 0.23 ± 0.05 µm and b = 0.09 ± 0.02 µm) along the fiber axis were measured to have a periodic distance of around 2 ± 0.2 µm. The lamellar periodic distance mentioned in the literature for crystalline PLA is in the range of nanometers [43], unlike the values reported in this study. Since this degradation study was limited to only 25 days, a further decrease in the periodicity of the pores could be anticipated as more pores are likely to form over prolonged degradation times. Additionally, the two surface plots in the right portion of Figure 9 accentuate the periodicity in the pores as black depressions and white peaks. For the crystalline regions, the two perpendicular directions of the lamellar structure are much greater than the dimension along the chain axis. This results in media diffusion and, due to its open structure and fewer intermolecular interactions, to the onset of degradation in the non-crystalline region. The formation of elliptical pores upon degradation indicates that there is an asymmetric interaction between the two regions which is lower than the intermolecular interaction in the non-crystalline region. This can be attributed to partial orientation in the non-crystalline region while drawing of PLA fibers during the manufacturing stage [37,42]. These observations also demonstrate the importance of including consideration of the processing conditions when developing structure-property relationships for fibers [34]. This is particularly true for high performance applications, such as biomedical uses that mimic the performance requirements in the human body, and for predicting efficient and complete end-of-life degradation or resorption of fibrous materials. 

In contrast to the observations of surface erosion in these degraded PLA fibers, studies in the literature report bulk erosion, including comparable systems such as molded PLA structures. This is attributed to the fact that diffusion of media into the bulk is much faster compared to the loss of the degraded polymer from the bulk of the sample. The media diffusing into the PLA bulk has a much smaller size compared to the degraded oligomers from the bulk of the polymer. Secondly, undrawn films and precast samples lack inherent orientation, unlike fibers that experience stress induced orientation while spinning and drawing which results in the fringed lamellar model with higher crystallinity for fibers (Figure 9). Additionally, although fibers are thin (approximately 10–15 µm in diameter), their semi-crystalline structure hinders the diffusion of external media into the bulk, leading to surface erosion.

In addition, the decrease in mechanical properties during the initial period of degradation (Figure 2) can be explained by the fact that degradation leads to a decrease in molecular weight and an increasing percentage of smaller sized polymer chains. These shorter chains can easily slide past each other initializing a plasticizing effect. Such chain breakage was observed at the molecular level through MD simulations, and with the single and 20 chain systems the chemical degradation was enhanced in alkaline conditions by forming lower molecular weight products and a reduced fraction of the original higher molecular weight PLA chains. Experimentally, k calculated from the molecular weight measurements is reported to be 0.05–0.07 days^−1^ [37]. To compare the rates calculated for PLA degradation under alkaline conditions due to the different timescales for the MD simulations and the experiments, the superposition principle was used to scale the rate constants (k_s_) as represented in Equation (10),
(10)$$k_{s}=k_{7.4}*\log\left[\frac{t_{s}\left({\mathrm{pH}}_{S}\right)}{t\left(7.4\right)}\right]$$
where k_7.4_ is the rate that has been reported experimentally for PLA degradation as measured by calculating molecular weights using GPC for neutral conditions at pH 7.4, t is the degradation time at neutral pH as reported in different experimental studies (40–60 days), and t_s_ is the simulation time (0.5 ns) at the simulation pH (pH_s_). Using this equation, k_s_ is calculated to be 1.70–2.77 ns^−1^ for pH values ranging from 10–14. This is close to the rate constant for degradation observed at extreme alkaline conditions in the presence of 100 OH^−^ ions and k of 2.43 ns^−1^ (Figure 6b). In other simulated alkaline conditions however, lower rates were observed to those calculated using Equation (10), mainly because of the formation of reversible intermediates that reform original chains at the end of the degradation process. Similarly, in acidic and neutral conditions, barely any degradation is observed at all with k values approaching 0. Potential reasons for this result is due to the intermediates being formed, and the length of the simulation time being insufficient to observe the degradation in these conditions. Lastly, the linearity of the slopes for all of the degradation conditions confirms that only one degradation mechanism occurs through hydrolysis rather that autocatalysis, which is reported to be caused by the acetic acid that results in chain backbiting.

Furthermore, the presence of OH^−^ enhances the rate of degradation for PLA, triggering the nucleophilic attack at electron deficient carbonyl carbons. This is different from acid catalyzed reactions, where H^+^ reacts with water to form a hydronium ion, which is then attracted to negative carbonyl oxygens. The carbonyl carbon becomes electrophilic attracting water so as to initiate hydrolysis. However, due to the limited charge concentration to make carbonyls, the use of the simulation enabled a carbon negative alkali attack to be observed (Figure 5). This is validated by electronic density distributions as calculated using the NPA analysis and the Fukui reactivities (Figure 7), confirming that electron distributions in an aqueous environment for PLA fibers create conditions that are viable for hydrolytic degradation in alkali rather than acidic conditions [44,45].

Lastly, as mentioned previously, the Youngs modulus for all the pH conditions measured experimentally (Figure 2) is initially observed to decrease, owing to the reduced molecular weight and increased plasticizing action of smaller molecular weight chains. However, during longer degradation periods the Youngs modulus was observed to increase by almost 28% after degradation for 25 days in alkaline conditions. This can be attributed firstly to an increase in crystallinity for all the samples after 25 days of degradation (Figure 4), as determined by the 16.4° peak which became narrower and sharper with increasing exposure time. In addition, the intensification and formation of new peaks at 28.8° and 33.3° due to degradation may be associated with an overall increase in sample stiffness. The higher ratio for peak heights and lower ratio for peak areas for the 16.4° peak with respect to the 28.8° peak confirms that degradation at pH 10 forms narrower and sharper peaks and generates samples with the largest crystal size, as observed experimentally.

## 5. Conclusions

For PLA fibers, the hydrolytic degradation was observed to occur at an enhanced rate in alkaline conditions as compared to other pH values, with surface erosion being greatest at pH 10 after 25 days. There was minimal change observed in surface and performance properties of the PLA fibers at pH 7.4, pH 3 and pH 2. This is in contrast to the bulk erosion reported in most of the PLA-based literature for hydrolytic degradation, including molded PLA grafts. This is attributed to the higher crystallinity for the fiber samples due to the fiber spinning process, which accentuates the importance of considering processing conditions when evaluating the performance of polymer materials. The molecular level information from the DFT calculations indicate that the positive center created at carbonyl carbon due to electron withdrawing groups results in the nucleophilic attack site by hydroxide anions. The MD simulations also corroborated the higher rates of degradation at higher pH conditions. These results can be used to develop material informatic models for devising structure-processing-property relationships for PLA and other biopolymers, further advancing sustainable material design and development. Additionally, the results can be used to design PLA fibrous materials for specific chemical environments, not only for medical device applications, but also other applications, such as geotextile end-uses. These results can also be used to tune the PLA processing conditions for complete end-of-life chemical degradation and resorption.

## Figures and Tables

**Figure 1 molecules-26-07554-f001:**
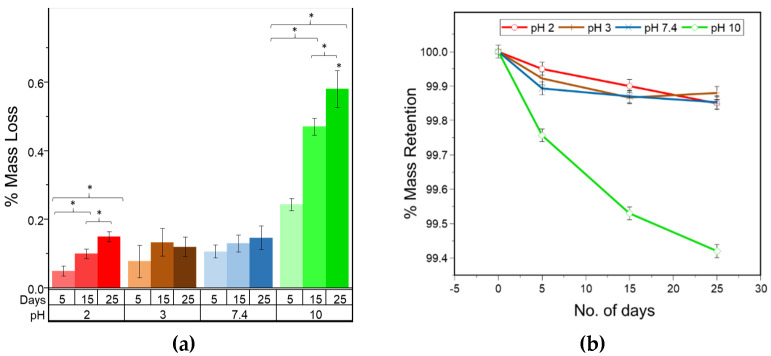
(**a**) % mass loss for different pH and degradation times, (* *p* < 0.05); (**b**) % mass retention for PLA fibers over time at different pH conditions.

**Figure 2 molecules-26-07554-f002:**
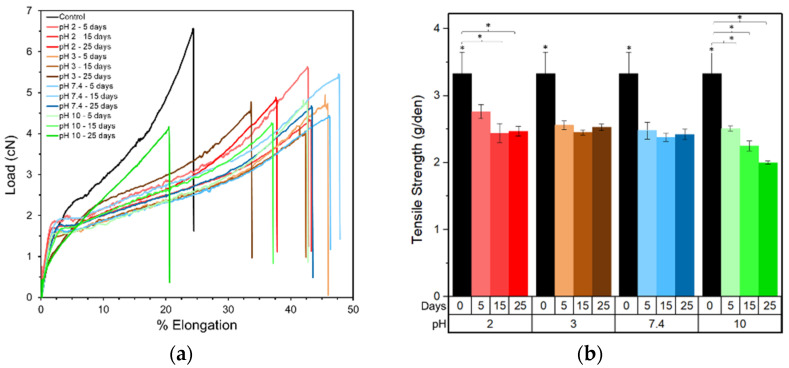
Tensile properties as a function of time and pH for the PLA fiber specimens. (**a**) Load-elongation plot; (**b**) tensile strength (g/den) (* *p* < 0.05); (**c**) % strength retention; and (**d**) Young’s modulus (g/den).

**Figure 3 molecules-26-07554-f003:**
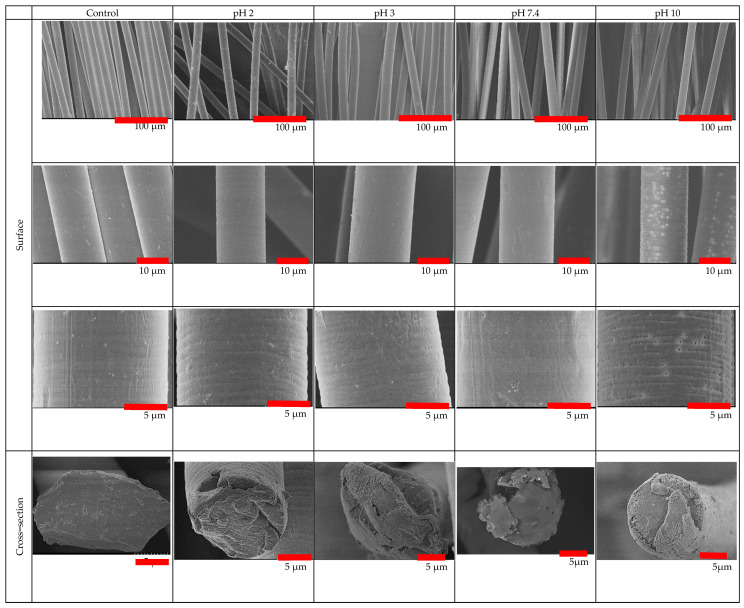
SEM images of PLA fibers after 25 days of exposure to different pH conditions.

**Figure 4 molecules-26-07554-f004:**
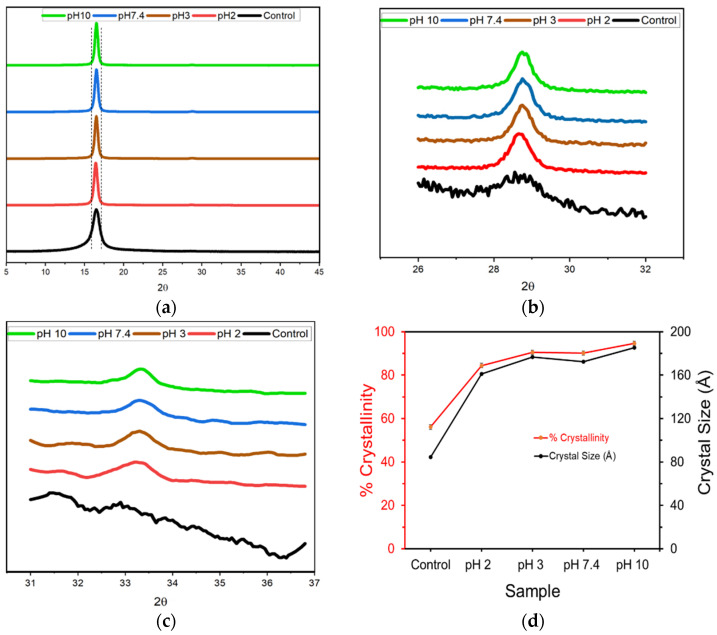
Wide angle XRD plots for the PLA fibers after 25 days degradation at pH 2, 3, 7.4 and 10 compared with the control sample (**a**) from 5° to 45°, (**b**) from 26° to 32°, (**c**) from 31° to 37°, and (**d**) the percent crystallinity (red circles, left axis) and crystal size (black triangles, right axis) calculated from the XRD plot by analyzing the highest intensity peaks.

**Figure 5 molecules-26-07554-f005:**
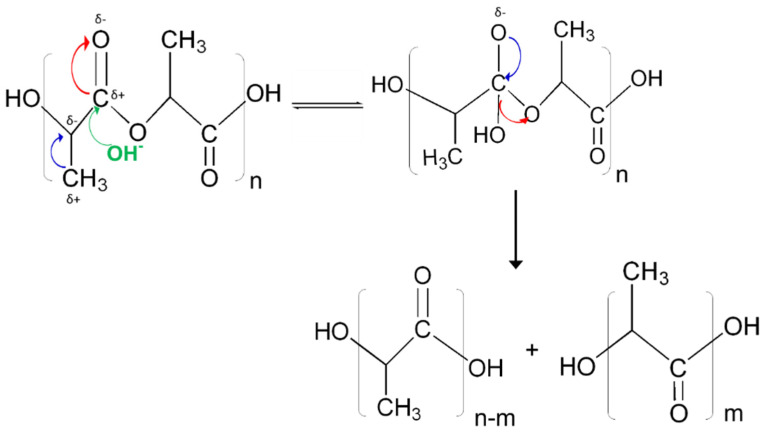
Schematic of the alkali-assisted hydrolytic degradation mechanism that was observed in the MD simulations of PLA chains.

**Figure 6 molecules-26-07554-f006:**
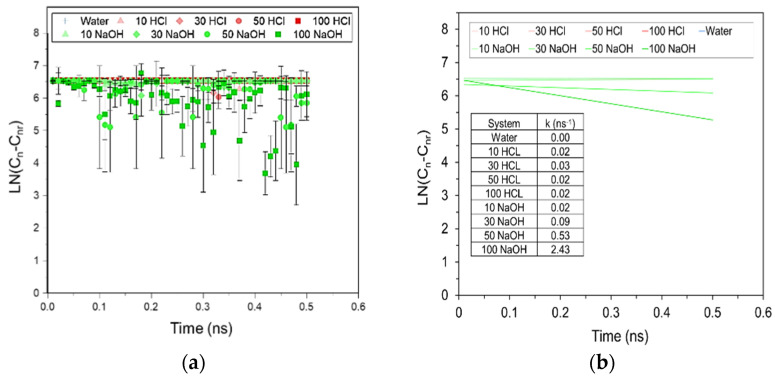
Analysis of the MD simulation trajectory of 20 chains of PLA in water as a function of time and increasing acidic (HCl, red) and alkaline (NaOH, green) environments. (**a**) The change in the carbon molecular weight of the PLA chains (LN(C_n_-C_nr_)) versus time, (**b**) Linear plots of the average values of LN(C_n_-C_nr_) vs time, where the calculated slopes that correspond to the rate constants (k, in ns^−1^) are given in the table inset.

**Figure 7 molecules-26-07554-f007:**
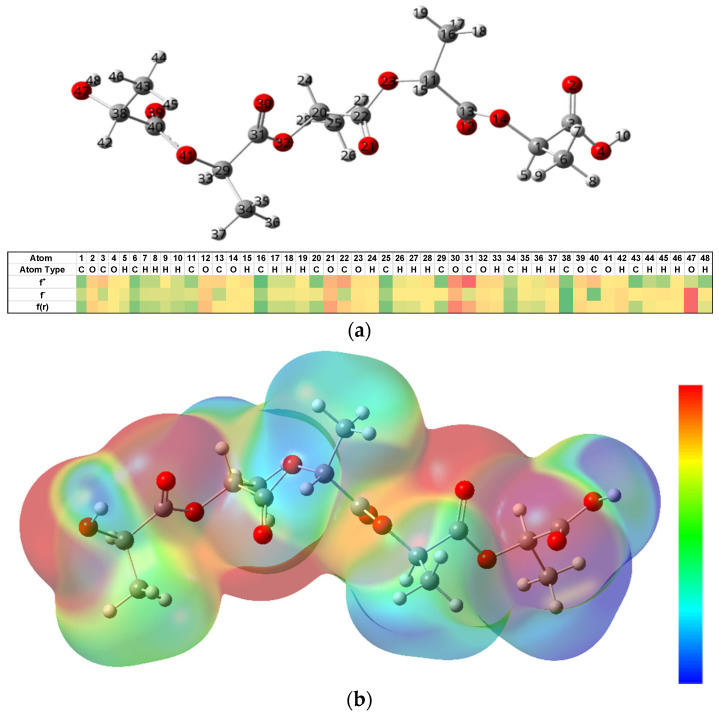
Results from the DFT calculations on a 5 repeat unit PLA chain. (**a**) Optimized chemical structure and a heat map of the Fukui atomic reactivity distributions, where red indicates the most reactive sites and green is the least reactive; (**b**) 3D contour representing the electrostatic potential surface, where red is more negative and blue is more positive.

**Figure 8 molecules-26-07554-f008:**
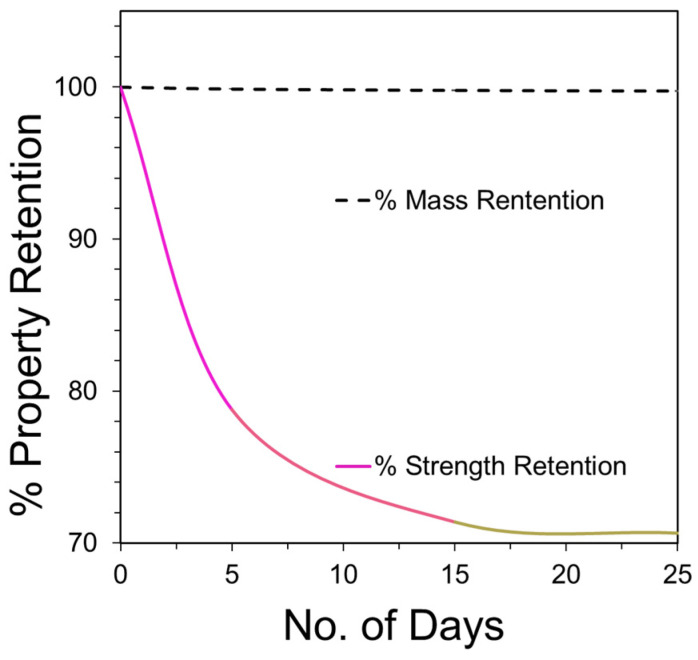
PLA fiber property retention at pH 10 in terms of % mass, and % strength retained over the degradation period of 25 days.

**Figure 9 molecules-26-07554-f009:**
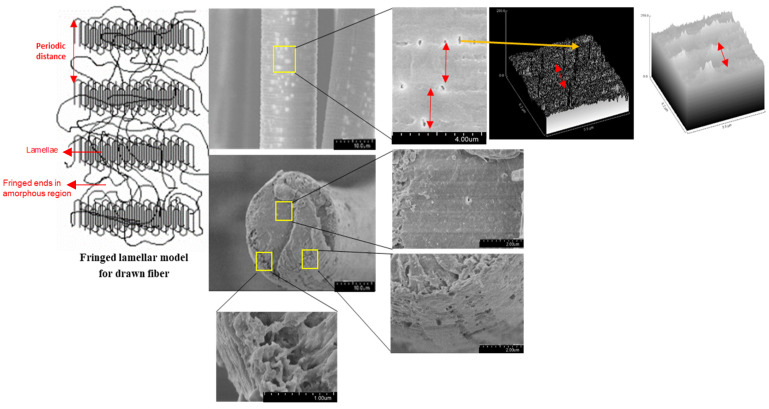
Left: lamellar model of the crystalline and amorphous regions for drawn fibers. Middle: magnified SEM images of the surface (top) and cross-section (bottom) of the degraded PLA fibers, which illustrate surface erosion that is likely occurring at the non-crystalline sites in the fringes. Right: Surface plots generated from Image J software, where the red arrows illustrate the periodicity.

## Data Availability

The data that support the findings of this study are available from the corresponding author upon reasonable request.

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
