# Peer review of "Hydrolytic Degradation of Polylactic Acid Fibers as a Function of pH and Exposure Time"

_molecules, 2021, doi:10.3390/molecules26247554_

Round 1

Reviewer 1 Report

The well prepared manuscript on the degradation mechanism of PLA studied with the experiment and simulation methods presents interesting results, which is recommended to be published after a minior revision. The suggestions are given below:

  1. The mass change of PLA fiber after 25 days is too slight to demonstrate the effect of degradation on mass, even in the high pH environment. Please improve the analysis to support the result.
  2. Part 2.5 is the degradation MD simulations. The topic should be revised to make it more appropriate.
  3. As mentioned in the MD simulation part, -COOH ends are formed by the break of ester bonds, while the molecular weight decreases. This can be proved by several well designed experiments such as titration and GPC. Please add the experiment to help the authors understanding.

Author Response

Point 1:  The mass change of PLA fiber after 25 days is too slight to demonstrate the effect of degradation on mass, even in the high pH environment. Please improve the analysis to support the result.

Response 1: Mass change is the last change observed after strength, change in surface topography and molecular weight. Thus, it is correctly highlighted by reviewer the mass loss isn’t significant after 25 days. If the degradation is continued for long term the mass loss might be significant attributed to the PLA properties.

However, this paper focused to develop a methodology to determine hydrolytic degradation of PLA using experiments and simulations. By observing trends and comparing the findings at macroscale and molecular level the overall results and outcome serve the purpose of study. Once this methodology is developed, it will be used to evaluate other less investigated polymers in the long term for spectra of applications

Point 2:  Part 2.5 is the degradation MD simulations. The topic should be revised to make it more appropriate.

Response 2: In the paper part 2.5 is computational analysis comprising of MD simulations and DFT calculations comprehensively describing the degradation of PLA at molecular level. The trends in the simulated pHs are similar to degradation trends from the experiments. MD simulations provide us the information on reaction mechanism, degradation products and reaction rates. The reaction mechanisms are thereafter validated using DFT calculations, confirming that the positive center created at carbonyl carbon due to electron withdrawing groups results in the nucleophilic attack site by hydroxide anions.

Point 3:  As mentioned in the MD simulation part, -COOH ends are formed by the break of ester bonds, while the molecular weight decreases. This can be proved by several well designed experiments such as titration and GPC. Please add the experiment to help the authors understanding.

Response 3: No GPC and titration were done as the results from these analyses are well understood from literature for PLA polymers, being widely investigated. Leveraging the reported data, we wanted to use simulations to obtain molecular level understanding and compare it with material performances at macro scale for PLA hydrolytic degradation. As demonstrated from this research can be used to investigate new polymers (in plan for us) where intensive testing to validate the end groups for those observed in simulation is being pursued.

Thank You

Reviewer 2 Report

In the reviewer’s opinion, this manuscript is well prepared and contains sufficient technical contribution. Before the final decision is to be made, the reviewer recommends the authors to consider some minor comments as listed in the following:

  1. Abstract, line 21: Quantify the rate of degradation in terms of percentage.
  2. Line 139: Reference [33] should be placed immediately after “Xu et al.”. Also, it is unnecessary to indicate “Lebo” because the first name is not required.
  3. Materials and Methods should be brought before the section of Results (immediately after Introduction).
  4. What is the reason to choose the upper limit as 25 days?
  5. Line 152: Should it be “pH 2, 10%”?
  6. Figure 3: The magnification of the micrographs is not clear. Based on the reviewer’s observation, the magnification for each row of micrographs is different. Please add the corresponding scale for each row of them.
  7. Line 200: It should be Figure 4.
  8. Line 233: [38] should be placed after “equations (4) and (5)”.
  9. Line 234: Is there a mistake in the symbol “rho”? It appears differently on the pdf version that I receive.
  10. Line 250: Capitalize “F” for the “figure”.
  11. Figure 5(b) – some lines are not seen. Is it because of overlapping?

Author Response

Thank You
